# Strain variation in *Bacillus cereus* biofilms and their susceptibility to extracellular matrix-degrading enzymes

Eun Seob Lim[1,2], Seung-Youb Baek[2], Taeyoung Oh[2], Minseon Koo[1,3], Joo Young Lee[2,4], Hyun Jung Kim[1,2]*, Joo-Sung Kim[1,2]*

**1** Department of Food Biotechnology, Korea University of Science and Technology, Yuseong-gu, Daejeon, Republic of Korea, **2** Research Group of Consumer Safety, Research Division of Strategic Food Technology, Korea Food Research Institute, Wanju-gun, Jeollabuk-do, Republic of Korea, **3** Food Analysis Center, Korea Food Research Institute, Wanju-gun, Jeollabuk-do, Republic of Korea, **4** Department of Food Science and Biotechnology, Sungkyunkwan University, Jangan-gu, Suwon, Republic of Korea

* hjkim@kfri.re.kr (HJK); jskim@kfri.re.kr (JSK)

**Data Availability Statement:** All relevant data are within the paper and S1 Fig.

## Abstract

*Bacillus cereus* is a foodborne pathogen and can form biofilms on food contact surfaces, which causes food hygiene problems. While it is necessary to understand strain-dependent variation to effectively control these biofilms, strain-to-strain variation in the structure of *B. cereus* biofilms is poorly understood. In this study, *B. cereus* strains from tatsoi (BC4, BC10, and BC72) and the ATCC 10987 reference strain were incubated at 30˚C to form biofilms in the presence of the extracellular matrix-degrading enzymes DNase I, proteinase K, dispase II, cellulase, amyloglucosidase, and α-amylase to assess the susceptibility to these enzymes. The four strains exhibited four different patterns in terms of biofilm susceptibility to the enzymes as well as morphology of surface-attached biofilms or suspended cell aggregates. DNase I inhibited the biofilm formation of strains ATCC 10987 and BC4 but not of strains BC10 and BC72. This result suggests that some strains may not have extracellular DNA, or their extracellular DNA may be protected in their biofilms. In addition, the strains exhibited different patterns of susceptibility to protein- and carbohydrate-degrading enzymes. While other strains were resistant, strains ATCC 10987 and BC4 were susceptible to cellulase, suggesting that cellulose or its similar polysaccharides may exist and play an essential role in their biofilm formation. Our compositional and imaging analyses of strains ATCC 10987 and BC4 suggested that the physicochemical properties of their biofilms are distinct, as calculated by the carbohydrate to protein ratio. Taken together, our study suggests that the extracellular matrix of *B. cereus* biofilms may be highly diverse and provides insight into the diverse mechanisms of biofilm formation among *B. cereus* strains.

## Introduction

*Bacillus cereus* is a gram-positive and spore-forming bacterium that can cause foodborne illness. *B. cereus* is often found in agricultural products and in their cultivation environments,

**Funding:** This research was supported by Main Research Program (E0192101-02) of the Korea Food Research Institute (KFRI) funded by the Ministry of Science and ICT in South Korea. The funders had no role in study design, data collection and analysis, decision to publish, or preparation of the manuscript.

**Competing interests:** The authors have declared that no competing interests exist.

including soils and irrigation water [1, 2]. The common presence of *B. cereus* in soil makes it reasonable that foodborne illnesses caused by *B. cereus* are associated with a broad spectrum of foods, not only foods of plant origin but also meat and dairy products [3].

Biofilms are biologically active matrixes of extracellular polymeric substances (EPS) secreted by microorganisms, in which the microbial cells are embedded. This form of life is often advantageous for surviving harsh environmental conditions, such as disinfectants [4]. Most foodborne pathogenic bacteria can form biofilms, and numerous studies have shown that *B. cereus* can form biofilms on food contact surfaces or in food-processing environments, which is a clear concern for food safety [5–8].

EPS are usually composed of carbohydrates, proteins, extracellular DNA, and lipids [9]. Although these compounds are commonly found in bacterial biofilms, their structures and functions are highly diverse depending on the bacterial species [9]. Furthermore, intraspecies variation exists in EPS composition [10].

The biofilm structure formed by *B. cereus* is poorly understood [11]. Particularly, strain-dependent variation in *B. cereus* biofilms in terms of EPS structures and compositions remains largely unknown. *B. cereus* is genetically diverse [11–14], and their biofilm-forming abilities are highly variable [11]. It is important to understand the strain-dependent variation in *B. cereus* biofilms to effectively control them.

Enzymes have emerged as one alternative strategy for controlling biofilms in the food industry because they are able to degrade essential components of the biofilm matrix, such as extracellular DNA, proteins, and carbohydrates [15]. In addition, enzymes can be useful to study the compositions of the EPS in bacterial biofilms by investigating their inhibitory effect on biofilm formation [16, 17]. Therefore, we studied six kinds of EPS-degrading enzymes, DNase I, proteinase K, dispase II, cellulase, amyloglucosidase, and α-amylase to characterize the biofilms of different *B. cereus* strains. In this study, we used *B. cereus* strains that showed strong biofilm-forming abilities isolated from tatsoi from our previous study [18] as well as the ATCC 10987 reference strain.

## Materials and methods

### Biofilm-forming *B. cereus* isolates

A total of 73 *B. cereus* isolates comprised 25 *B. cereus* isolates obtained from microgreen samples, 12 *B. cereus* isolates obtained from seed samples and 36 *B. cereus* isolates obtained from water, nutrients, and soil samples from farms as described in our previous study [18]. To isolate *B. cereus* and *B. thuringiensis*, polymerase chain reaction (PCR) amplification of the crystal (*cry*) gene was carried out for the biochemically identified *B. cereus* group isolates [19]. Biochemical identification for the *B. cereus* group was conducted by using the Vitek-II system with the BCL card (bioM´erieux, Inc., Marcy l'Etoile, France), according to the manufacturer's directions. The specific primer pair K3 (5′–GCTGTGACACGAAGGATATAGCCAC–3′) and K5 (5′–AGGACCAGGATTTACAGGAGG–3′) was used for the identification of the *cry* gene (1,600 to 1,700 bp) for *B. thuringiensis*. Template DNA was preheated at 94˚C for 7 min. Then, the DNA was denatured at 94˚C for 60 s and annealed to primers at 58˚C for 90 s; and the PCR products were extended at 72˚C for 60 s for 30 cycles for the *cry* gene [19, 20].

All the isolates were examined to determine their biofilm-forming abilities in a microtiter 96-well plate using a crystal violet assay, as in our previous report (under review) and according to Zhu et al. [21]. ATCC 10987 was used as a reference strain for biofilm formation. In brief, a total of 200 μl of Tryptic Soy Broth (TSB) (Merck, darmstadt, Germany) containing 2 μl of a 0.5 McFarland *B. cereus* group suspension was added to each well of the plate and incubated at 30˚C for 1 and 2 days without changing the medium according to the procedure

developed by Zhu et al. [21]. After the incubation period, biofilms were stained with 0.5% crystal violet solution and the absorbance at 570 nm was analyzed. The biofilm-forming ability of each strain was categorized as no, weak, moderate, or strong biofilm formation on the basis of the cutoff optical density (ODc) of the biofilms. The ODc for the microtiter plate test was defined as three standard deviations above the mean OD of the negative control: no biofilm formation (ODs ≤ ODc), weak biofilm formation (ODc < ODs ≤ 2× ODc), moderate biofilm formation (2× ODc < ODs ≤ 4× ODc), and strong biofilm formation (4× ODc < ODs) [22].

Among the *B. cereus* isolates, ones with the highest biofilm formation abilities were selected from each category, i.e., strain BC4 from microgreen samples, BC10 from seeds of microgreens and BC72 from water samples.

## Preparation of cell suspension

Each strain was stored in 30% glycerol at −70˚C and was separately cultured on MYP agar at 30˚C for 24 h. The single colonies were inoculated in TSB and incubated at 30˚C for 16–18 h in a shaking incubator. The overnight culture was diluted to approximately $10^7$ CFU/mL in TSB to prepare cell suspensions for biofilm study.

## Enzymes used in susceptibility test

The enzymes used in this study were: DNaseI (Thermo Scientific™, Lithuania, EU), the 2 proteases of proteinase K (Qiagen, Hilden, Germany) and dispaseII (Sigma-Aldrich, St. Louis, MO, USA) and the 3 polysaccharidases of cellulase (Duchefa Biochemie, Haarlem, the Netherlands), amyloglucosidase (Sigma-Aldrich, St. Louis, MO, USA), and α-amylase (Sigma-Aldrich, St. Louis, MO, USA). The final concentrations of the enzymes used in this study were 0.1% for DNaseI, 1% for proteinase K, amyloglucosidase and α-amylase, and 20 mg/ml for dispaseII and cellulase.

## Biofilm inhibition or removal using enzymes

The previous protocol was used with modification to study biofilm inhibition or removal using enzymes [23]. To study the inhibition of biofilm formation, cell suspensions in TSB prepared as above were combined with each enzyme in 200 μl in 96-well polystyrene microtiter plates (SPL, Pocheon, Korea), and the outermost wells were filled with deionized water to prevent evaporation. The plates were incubated at 30˚C for 48 h to facilitate biofilm formation. After incubation, the culture medium was carefully removed, and the wells were washed once with PBS. Then, 200 μl of crystal violet solution diluted to 1% (bioWORLD, Dublin, Ohio, USA) was added for staining biofilm matrix, and the microtiter plate was incubated for 30 min at room temperature (RT). After washing with PBS thrice, 200 μl of absolute ethanol (JT Baker, MA, USA) was added and incubated for 15 min at RT for destaining. From the destained solution, 100 μl was transferred to a new 96-well plate, and the absorbance was measured at 595 nm using a microplate reader (Infinite® 200 PRO NanoQuant, Tecan, Männedorf, Switzerland). When the measurement was outside the detectable range in a sample that exhibited strong biofilm formation, the destained solution was diluted 10-fold in absolute ethanol and measured for absorbance. The measurement of 10-fold diluted samples was multiplied by 10 to obtain the measurement for the un-diluted samples. The final absorbance was calculated by subtracting the average value of the wells with no inoculation (TSB only) from the value of each well with microbial inoculation. To study the enzyme-mediated removal of preformed biofilms, the TSB spent during biofilm formation was replaced with fresh TSB containing each enzyme and incubated at 37˚C for 1 h. After the wells were washed once with PBS, the same procedure was followed for crystal violet staining, washing, elution, and absorbance measurement.

## Confocal Laser Scanning Microscopy (CLSM) images

The cell suspension was prepared as aforementioned. Then, it was incubated with enzymes as described above on a 96 well plate with glass-like polymer coverslip on the bottom (μ-plate 96 well black, ibidi, Gräfelfing, Germany). After incubation, each well was washed 4 times with 200 μl filter-sterilized deionized water. The bottom-surface attached biofilms were stained with fluorescent dyes, SYTO®9 and propidium iodide included in Filmtracer™ LIVE/DEAD™ biofilm viability kit (Life Technologies, Eugene, OR, USA), following the manufacturer's protocol. CLSM images were taken using LSM 880 with airyscan (Carl Zeiss, Oberkochen, Germany) at 200 times magnification. The excitation wavelength for SYTO®9 green fluorescent nucleic acid stain and propidium iodide was 488 nm and 561 nm, respectively, and the emission wavelength ranges were 499–535 nm and 568–712 nm for SYTO®9 green fluorescent nucleic acid stain and propidium iodide, respectively. For differential imaging of protein, carbohydrate, and cells in biofilms, a previous protocol was basically followed [24]. The biofilms formed on a 96 well plate as described above were washed 4 times with 200 μl of sterile PBS for each time. SYTO 63 dye (Life Technologies) diluted to 20 μM was added at 200 μl to each well and incubated at RT for 30 min in the dark. Then, the wells were washed with sterile PBS for 1 min. FITC solution (0.1 mg/ml) (Sigma-Aldrich) prepared in 0.1 M sodium bicarbonate solution (pH 8.5–9.0) was added at 200 μl to each well and incubated at RT for 1 h in the dark. After the wells were washed in sterile PBS for 1 min, concanavalin A, tetramethylrhodamine conjugate (Life Technologies) diluted to 0.1 mg/ml in 0.1 M sodium bicarbonate solution (pH 8.5–9.0) was added at 200 μl to each well and incubated at RT for 30 min in the dark. Finally, the wells were washed with filter-sterilized deionized water 2 times with 1 min for each time. The protein, carbohydrate, and cells in bottom-surface attached biofilms were differentially visualized with LSM 880 with airyscan (Carl Zeiss) at the excitation wavelength/emission wavelength ranges of 488 nm/500-535 nm, 561 nm/571-615 nm, and 633 nm/650-758 nm, respectively.

## Biofilm morphology

To assess the diversity in biofilm morphology of the tested strains, biofilms were formed using the method described by Wu and Xi with some modifications [25]. The bacterial cells of each strain were incubated in 3 ml TSB in each well of a 6-well polystyrene microtiter plate (SPL) at 30˚C for 48 h. Images of the biofilms in the culture media were captured from above each well. To obtain images of polystyrene surface-attached biofilms, each well was washed thrice in 3 ml of PBS. Then, 3 ml of 1% crystal violet solution was added to each well and incubated for 30 min at RT. After washing with PBS 3 times, the images were captured by a commercial digital camera (EX2F, Samsung, Korea).

## Viable counts

After incubation of the bacterial cells, as in the biofilm morphology study, each well was washed thrice in 3 ml of PBS by adding the PBS against the sidewall with a pipette and removing it by tilting the plate at the corner. Then, the biofilm sample in each well was scraped off the wall and bottom surfaces of the plate and resuspended in 1 ml of sterile 0.85% saline. The resuspended samples were vortexed, serially diluted, and spread on TSA. The colonies grown on the TSA were enumerated after incubation at 30˚C for 48 h.

## Extraction, purification, and quantification of EPS

EPS were extracted from the bacterial biofilms by the method described by Eboigbodin and Biggs with some modifications [26]. As described above, a bacterial cell suspension ($10^7$ CFU/

mL) in TSB was prepared and incubated at 3 ml per well in a 6-well polystyrene microtiter plate (SPL) with two wells per strain under humidified conditions with deionized water, and the culture medium was carefully removed. Then, each well was washed thrice with 3 ml PBS, and the biofilm sample from each well was resuspended in 1 ml of sterile 0.85% NaCl solution by scraping the bottom and wall surfaces of the wells. The biofilm sample was transferred to a microcentrifuge tube, and centrifuged at $2,300 \times g$ for 15 min [26]. Bound EPS were extracted from the pellet, and free EPS were extracted from the supernatant.

To extract the bound EPS, the pellet was resuspended in 1 ml of a 1:1 solution of 0.85% NaCl and 2% EDTA then stagnantly incubated for 60 min at 4˚C. Then, the supernatant was harvested by centrifugation at 10,000 x $g$ at 4˚C for 30 min and filter-sterilized using a 0.45 μm pore size hydrophilic PVDF membrane (Millex®-HV 0.45 μm, Merck Millipore, Darmstadt, Germany).

To extract the free EPS, after centrifugation of the biofilm samples, the supernatant was filter-sterilized and precipitated with 1:3 volume ethanol and stored at -20˚C for 18 h. Then, the supernatant was removed after centrifugation at $10,000 \times g$ at 4˚C for 30 min. The pellets were dried and resuspended in 2 ml of a 1:1 volume solution of 0.85% NaCl and 2% EDTA.

The extracted EPS samples were analyzed to quantify the extracellular DNA (eDNA), protein, and carbohydrate. eDNA was purified from the samples by removing the proteins and RNA using MasterPure Gram Positive DNA Purification Kit (Epicentre, Madison, WI, USA), and the DNA concentration was measured using NanoVue Plus (GE Healthcare, Holliston, MA, USA). In the EPS samples, the protein was quantified by Pierce[TM] BCA Protein Assay Kit (Thermo Scientific[TM], Rockford, IL, USA) and the carbohydrate was quantified by Total Carbohydrate Assay Kit (Sigma-Aldrich, St. Louis, MO, USA).

### Statistical analysis

Statistically significant differences were determined by Tukey's Honest Significant Difference (HSD) test for multiple comparisons using Minitab 17 (Minitab Inc., State College, PA, USA). The null hypothesis was rejected when the *p*-value was less than 0.05 ($p < 0.05$).

## Results

### Susceptibility of surface-attached biofilms to extracellular polymeric substance (EPS)-degrading enzymes

The *B. cereus* isolates with strong biofilm formation abilities included 2 food isolates, strain BC4 (from microgreen leaves) and strain BC10 (from microgreen seeds), and 1 environmental isolate (strain BC72). Among them, strains BC4 and BC72 showed biofilm formation abilities similar to that of strain ATCC 10987 which was used as a reference strain (Fig 1A). To understand the correlation between biofilm mass and the number of biofilm-embedded viable cells, viable counts were conducted on the biofilm samples (Fig 1B). The viable counts did not necessarily correspond to the biofilm mass. The numbers of viable cells in strains ATCC 10987 and BC4 were similar and higher than that in strain BC10 ($p < 0.05$), and these numbers did correspond to the biomass (Fig 1). However, the viable count in strain BC72 was more than 10-fold lower (by 1.5 log cfu/cm$^2$) than that in strain BC4, although the biomass was comparable for both strains (Fig 1).

The biofilms of four strains, BC4, BC10, BC72, and the reference strain, ATCC 10987, were studied in terms of their susceptibility to 6 different enzymes that degrade DNA, carbohydrates, or proteins, which generally comprise the matrix of biofilms. The enzymes were added to the inoculum or to the preformed biofilms, incubated, and the biofilm mass was indirectly measured by the absorbance of crystal violet eluted from the stained biofilms. In general, the

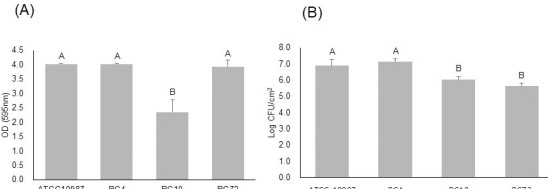

**Fig 1.** Biofilm-forming abilities (A) and viable counts of biofilm-embedded cells (B) of *Bacillus cereus* isolates from microgreens and their environmental samples. Strains BC4, BC10, and BC72 were isolated from microgreen leaves and seeds, and environmental water samples, respectively. The different letters above the bars indicate significant differences at $p < 0.05$ using Tukey's HSD.

biofilms were more effectively controlled when the enzymes were added to the inoculum than to the preformed biofilms (Fig 2). Although three strains, ATCC 10987, BC4, and BC72, were still strong biofilm formers in TSB, BC10 formed a weak biofilm in TSB. The four different strains showed different patterns of enzyme susceptibility (Fig 2). In the inhibition of biofilm formation, for example, the biofilm formation of ATCC 10987 was significantly reduced in the presence of DNase I, dispase II, and cellulase ($p<0.05$) (Fig 2A), while the biofilm formation of BC72 was significantly reduced by proteinase K, dispase II, and amyloglucosidase ($p<0.05$) (Fig 2D). Dispase II was effective against three strains except for BC10, and this enzyme was the most effective among the tested enzymes. In the degradation of the preformed biofilms, strains ATCC 10987, BC10, and BC72 were resistant to all the tested enzymes (Fig 2E, 2G and 2H), whereas strain BC4 was susceptible to all the tested enzymes except amyloglucosidase (Fig 2F). When the strains were incubated in the presence of the enzymes, some enzymes promoted biofilm formation in a strain-dependent manner (Fig 2A and 2C). Biofilm formation was enhanced in the presence of α-amylase for strain ATCC 10987 (Fig 2A) and in the presence of cellulase and amyloglucosidase for strain BC10 (Fig 2C).

To confirm the effects of EPS-degrading enzymes on biofilms, the biofilms were formed in the presence of each of 4 different enzymes and were stained with fluorescent dyes, SYTO®9 and propidium iodide, and the bottom-surface attached biofilms were visualized using CLSM

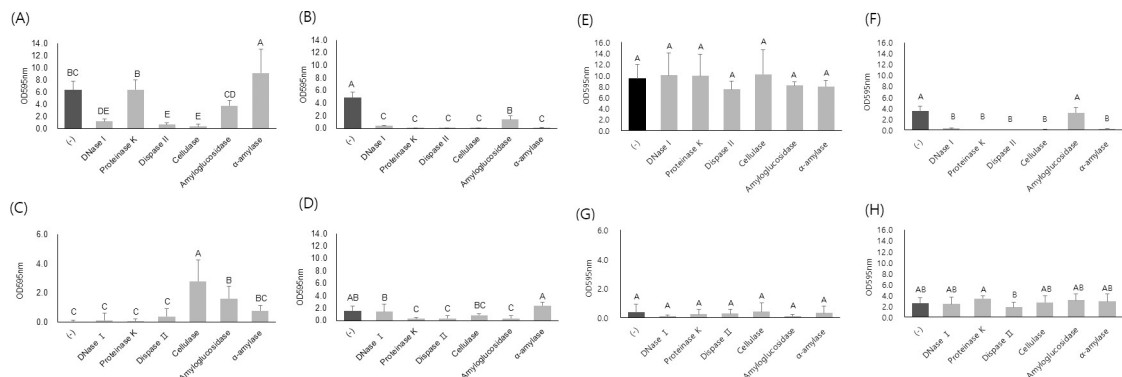

**Fig 2. Quantification of polystyrene surface-attached biofilms of *B. cereus* isolates treated with extracellular matrix-degrading enzymes.** Biofilms were formed in the presence of enzymes (A-D) or treated with enzymes after formation (E-H) on a polystyrene 96-well microtiter plate. The isolates, BC4 (B, F) and BC10 (C, G), were from microgreen leaves and seeds, respectively, and BC72 (D, H) was from water samples. These isolates were compared to a reference strain, *B. cereus* ATCC 10987 (A, E). Biofilms were formed in tryptic soy broth in the presence of each enzyme at 30°C for 48 h (A~D), or preformed biofilms were treated with each enzyme at 37°C for 1 h (E~H). Then, the biofilms were quantified by crystal violet staining. The data are from three independent experiments performed in triplicate. The different letters above the bars indicate significant differences at $p < 0.05$ using Tukey's HSD.

(Fig 3). Three different strains, ATCC 10987, BC4, BC72 had different patterns of biofilm formation and BC10 biofilm was rarely seen. All of the strains revealed different patterns of biofilm formation. Aggregative pattern was observed in ATCC 10987 while BC4 and BC72 revealed scattered and wire-like patterns, respectively. Most of the enzyme-susceptibility patterns were consistent with those of crystal violet staining assay (Fig 3).

Because DNase I susceptibility was variable depending on strains, we hypothesized that the susceptible strains contain eDNA in their EPS while the resistant strains do not. To test this hypothesis, DNA purified from EPS samples was run on agarose gels and visualized under UV light (Figs 4 and S1). DNA was found only in the bound EPS sample of strain ATCC 10987, confirming the presence of eDNA, while it was absent in the resistant strains, BC10 and BC72, confirming the absence of eDNA (Figs 4 and S1). However, DNA was also not found in the susceptible strain, BC4.

## Colony morphology

These four strains were inoculated onto MYP agar and the colony morphology was studied after 48 h incubation at 30°C (Fig 5). A typical pink color was observed for all the tested strains. Three strains, ATCC 10987, BC4, and BC72, formed round-shaped colonies while BC10 formed irregular colonies on MYP. Strains ATCC 10987 and BC72 had a very similar colony morphology with white surrounding zones, while strains BC4 and BC10 had red surrounding zones (Fig 5).

## Biofilm morphology

*B. cereus* strains were grown in TSB at 30°C for 48 h on a polystyrene microtiter plate. Then, we observed the formation of crystal violet-stained biofilms attached to the polystyrene surfaces or pellicles and flocs floating in the media (Fig 6). Strain ATCC 10987 formed a pellicle at

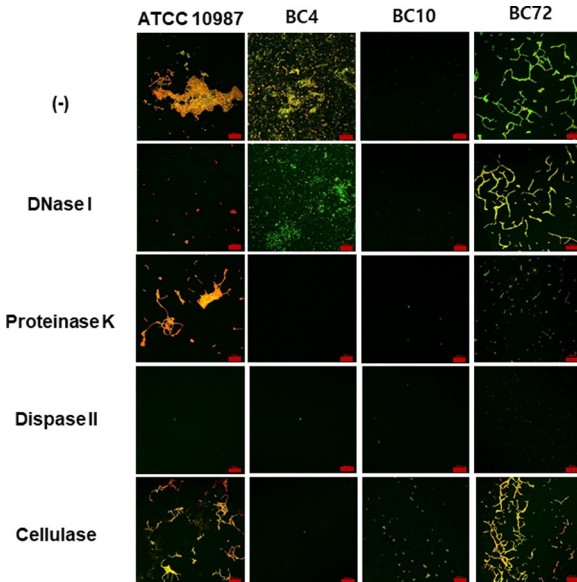

**Fig 3. CLSM images of bottom surface-attached biofilms of *B. cereus* isolates treated with extracellular matrix-degrading enzymes.** Biofilms were formed on a microplate with glass-like polymer coverslip on the bottom. The isolates BC4 and BC10 were from microgreen leaves and seeds, respectively, and BC72 was from water samples. These isolates were compared to a reference strain, *B. cereus* ATCC 10987. Biofilms were formed in tryptic soy broth in the presence of each enzyme at 30°C for 48 h and the bottom-surface attached biofilms were visualized using fluorescent dyes, SYTO®9 and propidium iodide in CLSM analysis. Scale bars represent 50 μm long.

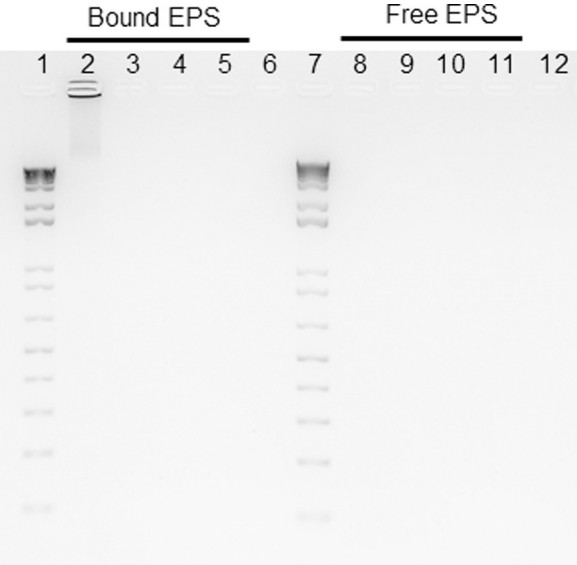

**Fig 4. DNA samples purified from extracellular matrices of the biofilms of *B. cereus* isolates.** The isolates, BC4 and BC10, were isolated from microgreen leaves and seeds, respectively, and BC72 was isolated from water samples. These isolates were compared to a reference strain, *B. cereus* ATCC 10987. DNA samples were analyzed by agarose gel electrophoresis and visualized under UV light. Bound EPS were loaded in lane 2 to 5, and free EPS were loaded in lane 8 to 11. Lanes 1 and 7, 1 Kb Plus DNA ladder; lanes 2 and 8, ATCC 10987; lanes 3 and 9, strain BC4; lanes 4 and 10, strain BC10; lanes 5 and 11, strain BC72; lanes 6 and 12, negative control (TE buffer only).

the air-liquid interface. After crystal violet-staining, a thick o-ring was clearly observed on the sidewall of the well at the interface, but very little was left on the bottom surface. However, strain BC4 showed a distinct pattern of biofilm formation (Fig 6). The majority of the biofilms formed a fluffy structure on the bottom of the well with little on the sidewall at the air-liquid interface. Different from strains ATCC 10987 and BC4, both strains BC10 and BC72 formed flocs in the suspensions. However, they also still formed submerged biofilms with a reduced amount of biofilm on the bottom surface compared to strain BC4, and that of strain BC10 was thicker than that of strain BC72 (Fig 6). Similar to BC4, both BC10 and BC72 formed little biofilm at the air-liquid interface on the sidewall.

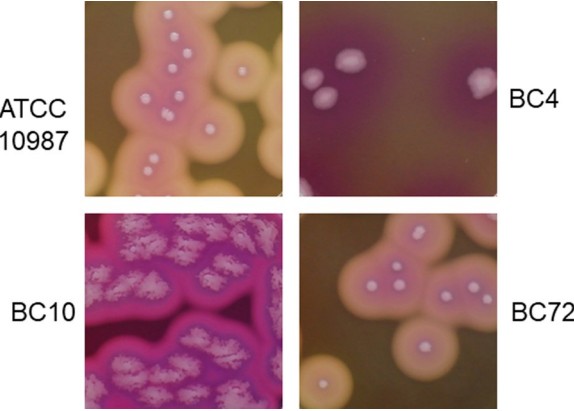

**Fig 5. Colony morphology of the *B. cereus* isolates from microgreen and water samples.** The isolates, BC4 and BC10, from microgreen leaves and seeds, respectively, and BC72 from water samples, were compared to a reference strain, *B. cereus* ATCC 10987. The isolates were grown on MYP agar after 48 h incubation at 30˚C.

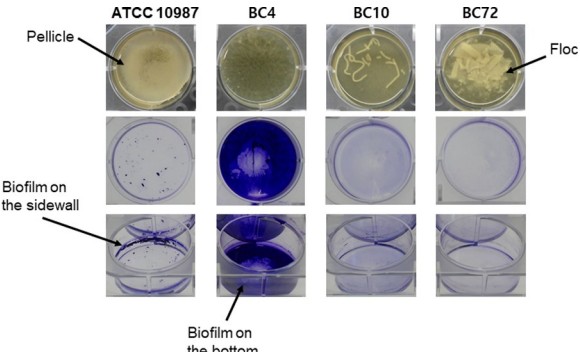

**Fig 6. Biofilm morphology of the *B. cereus* isolates from microgreen and water samples.** The isolates, BC4 and BC10, were isolated from microgreen leaves and seeds, respectively, and BC72 was isolated from water samples. The isolates were compared to a reference strain, ATCC 10987. The cells were incubated at 30°C for 48 h in 3 ml TSB on a 6-well polystyrene microtiter plate. Strain ATCC 10987 formed a pellicle on the surface of the liquid medium. The biofilm formation on the sidewall was clearly visible in the form of an o-ring, which corresponded to pellicle formation. Strain BC4 formed a biofilm mainly on the bottom of the well. In contrast to ATCC 10987 and BC4, which formed biofilms on the surfaces of the wells, strains BC10 and BC72 mainly formed flocs suspended in the media. The flocs of strain BC10 were filamentous, and the flocs of strain BC72 were flattened.

## EPS analysis

Strains ATCC 10987 and BC4 had comparable biofilm mass and viable counts, but distinct patterns in terms of susceptibility to EPS-targeting enzymes and colony and biofilm morphologies (Figs 1–3, 5 and 6). Therefore, excluding any effect of biofilm mass or viable cell number, the two strains were compared in terms of the quantity of major EPS components in the biofilms formed on the polystyrene surface. EPS were fractionated into free and cell-bound EPS based on their distance and attachment patterns to the cells [26]. The amount of eDNA, protein, and carbohydrate was analyzed because these are the major components of biofilms, including those formed by *B. cereus* [11]. The amounts of eDNA, protein, and carbohydrate were all higher in strain BC4 than strain ATCC 10987 (Fig 7). The relative amount of each component in the free and cell-bound EPS was compared. In strain ATCC 10987, the amounts of eDNA and carbohydrate were comparable between the free and cell-bound EPS (Fig 7A and 7C) while the amount of protein was much higher in the cell-bound EPS (~12-fold) than in the free EPS (Fig 7B). However, different from strain ATCC 10987, the amount of eDNA was approximately 3-fold higher, while the amount of carbohydrate was approximately 4-fold lower, in the cell-bound EPS than in the free EPS in strain BC4 (Fig 7A and 7C). In addition, the amount of protein was comparable (<2-fold) between the free and cell-bound EPS in strain BC4 (Fig 7B). The ratio of the amount of carbohydrate to protein in the EPS was analyzed because it is highly related to the physicochemical properties of biofilms [26]. This ratio was much higher in strain ATCC 10987 than strain BC4 (Fig 7D). In strain ATCC 10987, the ratio of carbohydrate to protein was 13-fold and 5-fold higher in the free and cell-bound EPS, respectively, than in strain BC4. Overall, carbohydrate was most common in the EPS, followed by protein and eDNA, for both strains ATCC 10987 and BC4 (Fig 7E). The protein ratio in the total EPS was 3-fold higher in strain BC4 than in strain ATCC 10987 (Fig 7E).

To study the distribution of EPS and cells in the biofilms, protein, carbohydrate, and cells were stained with FITC, tetramethylrhodamine conjugate of concanavalin A, and SYTO 63, respectively, and observed in CLSM analysis (Fig 8). In ATCC 10987 and BC72, no segregation of protein, carbohydrate, and cells was observed and they were found on the same locations. While the fluorescent intensities between protein and carbohydrate

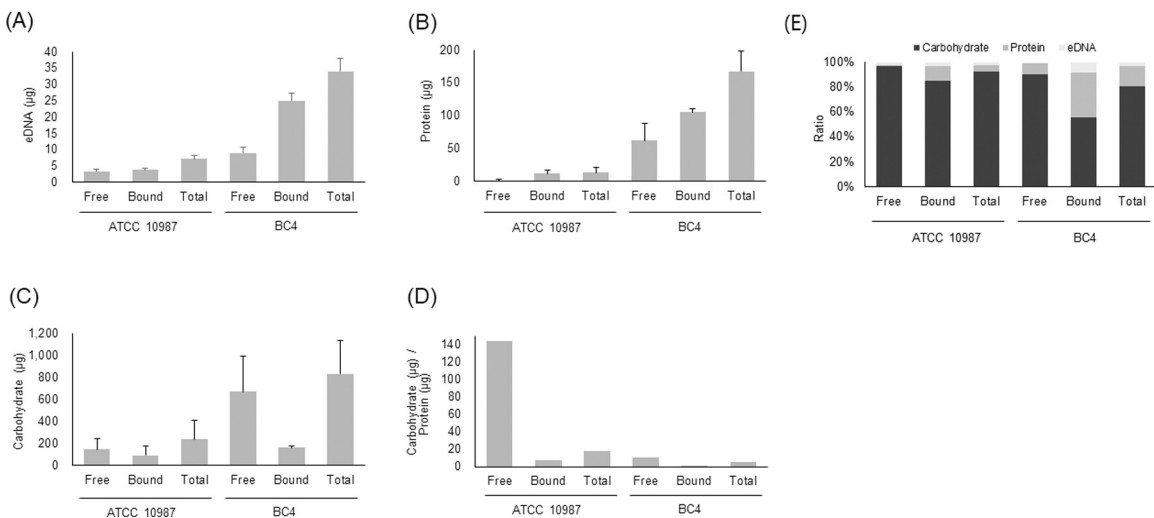

**Fig 7. The amounts of eDNA, protein, and carbohydrate of the extracellular matrix in the biofilms of *B. cereus* BC4 isolate and ATCC 10987.** The BC4 isolate was isolated from microgreen leaves. EPS of the polystyrene surface-attached biofilms were harvested from two wells of a 6-well polystyrene microtiter plate per strain and separated into cell-bound and free EPS. The amounts of (A) eDNA, (B) protein, and (C) carbohydrate in both the free- and cell-bound EPS were calculated. Total amounts of the EPS were calculated by adding the amount of cell-bound EPS to that of free EPS. The mass ratio of carbohydrate to protein in the extracellular matrix was largely different between ATCC 10987 and BC4 (D, E). Free, free EPS; Bound, cell-bound EPS; Total, free EPS plus cell-bound EPS. The data are based on 3 independent experiments.

were comparable in both ATCC 10987 and BC72, the relative intensity of protein was much higher in BC4, which is consistent with the increased ratio of protein in BC4 (Figs 7 and 8).

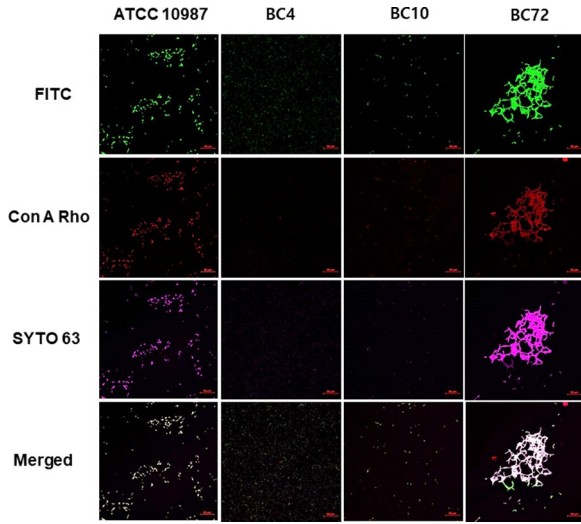

**Fig 8. Differential imaging analysis of protein, carbohydrate, and cells in the biofilms of *B. cereus* isolates.** Biofilms were formed on a microplate with glass-like polymer coverslip on the bottom. The isolates, BC4 and BC10, were isolated from microgreen leaves and seeds, respectively, and BC72 was isolated from water samples. These isolates were compared to a reference strain, ATCC 10987. Biofilms were formed in tryptic soy broth at 30°C for 48 h and the bottom-surface attached biofilms were visualized using CLSM. Protein, carbohydrate, and cells were stained with FITC, tetramethylrhodamine conjugate of concanavalin A (Con A Rho), and SYTO 63, respectively. Scale bars represent 50 μm long.

## Discussion

EPS are essential for forming biofilms, and both carbohydrate and protein components are generally known to play an essential role in the formation of EPS in *Bacillus* biofilms [27]. Our results show the large variation among *B. cereus* strains in terms of susceptibility to EPS-targeting enzymes (Figs 2 and 3). Initially, we performed this experiment to identify an optimal and common EPS-degrading enzyme to control *B. cereus* biofilms. Instead, we found a diverse pattern among *B. cereus* strains in terms of susceptibility to several EPS-degrading enzymes (Figs 2 and 3). We speculate that several factors can mainly affect the pattern of enzyme-susceptibility. First, the presence or absence of a certain EPS constituent affects the susceptibility to the corresponding enzymes [17]. For example, the presence or absence of eDNA made a large difference in the susceptibility of *Listeria monocytogenes* biofilms to DNase I treatment [17]. Second, the target constituent does not play an essential role in biofilm formation. Third, the target constituent is somehow protected from enzyme digestion. For example, the exogenous EPS-degrading enzymes could be degraded by the proteolytic enzymes secreted by microorganisms in biofilms and serve as a nutrient [28]. Fourth, exogenous EPS-degrading enzymes may degrade the secreted enzymes or molecules necessary for the process of biofilm detachment [29]. The patterns of resistance or susceptibility to the test enzymes may be mainly determined by these factors, and the third or fourth factor may have contributed to the unusually increased patterns, such as that observed under cellulase treatment in strain BC10 (Fig 2C). In addition to the distinct patterns of enzyme susceptibility, our data also reveal the distinct biofilm and colony morphologies of these strains (Figs 3, 5 and 6). In addition, compositional and imaging analyses of EPS showed the distinct patterns among the strains (Figs 7 and 8). Taken together, these data may suggest that surface-attached EPS of *B. cereus* biofilms are highly diverse in terms of composition, structure, function, and physicochemical properties.

eDNA released from bacterial cells by cell lysis or controlled secretion contributes to the adhesion on the surface, cell aggregations in the process of biofilm formation, the maintenance of structural integrity of biofilm as well as the genetic exchange by being horizontally transferred in bacterial community [30–32]. Therefore, it plays an important role in biofilm formation and is one of the essential constituents of the biofilms of many bacterial pathogens, including *B. cereus* [16, 32–35]. Vilain et al. [35] showed that *B. cereus* ATCC 14579 requires eDNA to form biofilms, and eDNA is found in the matrix of these biofilms. Consistently, eDNA should play an essential role in the biofilm formation of strains ATCC 10987 and BC4 (Fig 2), but no disruption of the preformed biofilm of strain ATCC 10987 was observed after treatment with DNase I; this result suggests that eDNA may play a minor or no role in the structural maintenance of this biofilm. Or, eDNA might exist in the complex with other EPS components and be protected from enzymatic digestion [36]. Furthermore, eDNA may not play an essential role in the biofilm formation of some environmental strains, such as BC72 (Fig 2). This finding may suggest that eDNA could be absent from the biofilms of some *B. cereus* strains in the environment.

Cellulose is one of the exopolysaccharides used in bacterial biofilm formation [37, 38]. However, it is still unknown if cellulose is essential in the biofilm formation of *B. cereus*. The cellulase-sensitive properties of strains ATCC 10987 and BC4 suggest that cellulose may play an essential role in their biofilm formation (Figs 2 and 3). On the other hand, a recent study by Whitfield et al. [39] suggests that the polysaccharide Pel is essential for the biofilm formation of *B. cereus* ATCC 10987. Considering that Pel is also cellulase-sensitive [40], the degradation of Pel by cellulase may have occurred in strain ATCC 10987, inhibiting its biofilm formation.

The biofilm formation of many bacterial species can be inhibited by proteinase K [41]. The patterns of resistance and susceptibility to proteinase K of strains ATCC 10987 and BC4, respectively, may suggest a significant difference in the role or structure of the proteins in biofilm formation or maintenance.

In EPS analysis of the strain ATCC 10987 biofilms, the amount of carbohydrate was much higher than that of protein in our study (Fig 7), while the amount of protein was slightly higher than that of carbohydrate in a previous study [42]. Although the reason underlying such a large difference remains unknown, different sources of EPS extraction (culture supernatant in the previous study vs. surface-attached biofilm in our study) could have contributed to the difference. This result may suggest that carbohydrates may play a significant role in surface attachment for biofilm formation (Fig 2A).

As shown in strain ATCC 10987, a pellicle formation at the air-liquid interface is a common biofilm morphology in *Bacillus* species such as *B. cereus* [5, 43] and *B. subtilis* [11, 27]. Wijman et al. [43] found that biofilms at the air-liquid interfaces are more abundant than submerged biofilms in *B. cereus*. In our study, however, submerged biofilm was a dominant phenotype in strain BC4, and it exhibited a distinct colony morphology and EPS composition compared to that of strain ATCC 10987 (Figs 5–7). In addition, strains BC10 and BC72 also formed submerged biofilms as well as flocs. This finding suggests that biofilm morphology may be highly diverse in environmental *B. cereus* strains and the population forming submerged biofilms could be substantial in the environment. A further study is warranted to understand the submerged biofilms and flocs in *B. cereus*.

In conclusion, this study shows that *B. cereus* biofilms are highly diverse depending on the strains and provides insight into the different mechanisms of biofilm formation. Such diversity may render the use of EPS-degrading enzymes helpful for characterizing *B. cereus* biofilms.

## Supporting information

**S1 Fig. DNA samples purified from extracellular matrices of the biofilms.**
(PDF)

## Author Contributions

**Conceptualization:** Joo-Sung Kim.

**Data curation:** Eun Seob Lim, Seung-Youb Baek, Taeyoung Oh.

**Formal analysis:** Eun Seob Lim, Joo Young Lee, Hyun Jung Kim.

**Investigation:** Eun Seob Lim, Seung-Youb Baek, Taeyoung Oh.

**Methodology:** Eun Seob Lim, Minseon Koo.

**Supervision:** Hyun Jung Kim.

**Writing – original draft:** Hyun Jung Kim, Joo-Sung Kim.

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
