## [Decision Letter · Decision Letter 0]

10 Feb 2021

PONE-D-21-00279

Strain variation in Bacillus cereus biofilms and their susceptibility to extracellular matrix-degrading enzymes

PLOS ONE

Dear Dr. Kim,

Thank you for submitting your manuscript to PLOS ONE. After careful consideration, we feel that it has merit but does not fully meet PLOS ONE’s publication criteria as it currently stands. Therefore, we invite you to submit a revised version of the manuscript that addresses the points raised during the review process.

We look forward to receiving your revised manuscript.

Kind regards,

Muhammad Hussnain Siddique

Academic Editor

PLOS ONE

Journal Requirements:

Reviewers' comments:

Reviewer's Responses to Questions

**Comments to the Author**

1. Is the manuscript technically sound, and do the data support the conclusions?

Reviewer #1: Partly

Reviewer #2: No

2. Has the statistical analysis been performed appropriately and rigorously? 

Reviewer #1: Yes

Reviewer #2: Yes

3. Have the authors made all data underlying the findings in their manuscript fully available?

Reviewer #1: Yes

Reviewer #2: Yes

4. Is the manuscript presented in an intelligible fashion and written in standard English?

Reviewer #1: Yes

Reviewer #2: No

5. Review Comments to the Author

Reviewer #1: In my opinion, the ms should be revised carefully before further consideration.

1. lines 89-93: This can be removed from M&M. It's not description of methods. More like a introduction.

2. lines 85: What's EPS-degrading enzymes here? The authors used DNAes, protease and cellulase?

3. Line 88: What's the definition of biofilm forming strain here? More description is needed.

4. Line 108: In my opinion, it's not biofilm of strain but only colony.

5. More advanced technology should be applied to understand the structure of colony (biofilm?) before and after enzyme applicaitn. Not only the appearacne. For example, TEM for the inner structure; Staining for protein distribution.

6. Why the extracellular DNA was secreted? Should explain this.

Reviewer #2: Line 38 "In this study, B. cereus strains from tatsoi and the ATCC 10987 reference strain" it's better to add codes of different strains used in this study like "B. cereus strains from tatsoi (BC4, BC10 and BC72)...........".

It is not clear how the isolated B. cereus strains were identified and confirmed as B. cereus. Chemical method or ribotyping? Please add results.

Line 123-124: Specify the staining solution?

Which device was used for imaging Biofilm morphology?

Study Biofilm inhibition lacks reference/s. Authors are advised to mention reference clearly which was used for this study.

Line 197 "and 1 environmental isolate (strain BC72), strains BC4 and" replace "," with full stop and make two sentences.

Line 377-389 "eDNA plays an important role in biofilm formation................ B. cereus strains in the environment". Figure 2 suggests that BC4 is susceptible to DNAse I, It is not clear why eDNA was not detected in biofilm of BC4? Suggest plausible reason.

It seems something was wrong with the data in figure 2C. The absorbance of crystal violet in the absence of degrading enzyme (-) (fig2C) is not comparable with the corresponding absorbance of crystal violet in fig 1A. Please comment.

Line 233-234: Why some enzymes promoted biofilm formation in a strain-dependent manner (Fig 2A and 2C).

Careful English editing is advised, help from a native-language expert or paid services shall be taken.

The word "in addition" in line 74 shall be replaced by an alternative word as it was previously mentioned in line 46 and then in line 82 again.

Line 76: Word “although” is not required.

Line 97-99 “Among the B. cereus .......... seeds and environmental samples” can be written as “Among the B. cereus isolates, ones with highest biofilm formation abilities were selected from each category i.e. strain BC4 from microgreen samples, BC10 from seeds of microgreens and BC72 form water samples were selected for current study”.

6. PLOS authors have the option to publish the peer review history of their article (what does this mean?). If published, this will include your full peer review and any attached files.

Reviewer #1: No

Reviewer #2: **Yes: **Muhammad Khurshid

---

## [Author Response · Author response to Decision Letter 0]

30 Apr 2021

PONE-D-21-00279

Strain variation in Bacillus cereus biofilms and their susceptibility to extracellular matrix-degrading enzymes

PLOS ONE

Dear Dr. Kim,

Thank you for submitting your manuscript to PLOS ONE. After careful consideration, we feel that it has merit but does not fully meet PLOS ONE’s publication criteria as it currently stands. Therefore, we invite you to submit a revised version of the manuscript that addresses the points raised during the review process.

We look forward to receiving your revised manuscript.

Kind regards,

Muhammad Hussnain Siddique

Academic Editor

PLOS ONE

Journal Requirements:

-> Thank you for your comments. We corrected our manuscript, especially the front page, to meet the PLoS One’s style requirements. For example, we removed the street addresses from the affiliations and the physical addresses from the corresponding authorship, and added the corresponding authors’ initials. 

 Thank you for the comments. We provide the correct grant numbers.

-> Thank you for your comments. We have only one gel picture in the manuscript. We converted the original, uncropped gel image to PDF file, named it ‘S1_raw_images’, and uploaded as supporting information to follow the instruction provided by PLoS One. 

-> Thank you for your comments. Now, we include the caption for our supporting information file, S1_raw_images under a newly created heading, ‘Supporting information’ as in line of 695-697 after revision in track changes. 

Reviewers' comments:

Reviewer's Responses to Questions

Comments to the Author

1. Is the manuscript technically sound, and do the data support the conclusions?

Reviewer #1: Partly

Reviewer #2: No

2. Has the statistical analysis been performed appropriately and rigorously? 

Reviewer #1: Yes

Reviewer #2: Yes

3. Have the authors made all data underlying the findings in their manuscript fully available?

Reviewer #1: Yes

Reviewer #2: Yes

4. Is the manuscript presented in an intelligible fashion and written in standard English?

Reviewer #1: Yes

Reviewer #2: No

 -> We had our manuscript edited by a paid editing service. We uploaded the editing certificate.

5. Review Comments to the Author

Reviewer #1: In my opinion, the ms should be revised carefully before further consideration.

1. lines 89-93: This can be removed from M&M. It's not description of methods. More like a introduction.

-> Thank you for your comments. According to your suggestion, a brief explanation on the B. cereus isolates was moved to introduction (line 89-90 after revision in track change), however more detail comments on the isolates, for example the numbers and sources of the isolates remain in materials and methods. 

2. lines 85: What's EPS-degrading enzymes here? The authors used DNAes, protease and cellulase?

-> Thank you for the good comments. We studied six different EPS-degrading enzymes, DNase I, proteinase K, dispase II, cellulase, amyloglucosidase, and α-amylase. We added this information in the text as in line 87-88 after revision in track change.

3. Line 88: What's the definition of biofilm forming strain here? More description is needed.

-> Thank you for your comments. The biofilm forming strain is defined now as in line 119-126 after revision in track change. Basically, we followed the previous protocol of Stepanović et al. (2000).

4. Line 108: In my opinion, it's not biofilm of strain but only colony.

-> Thank you for the good comments. As the reviewer pointed out, the overnight culture shouldn’t be a biofilm. It is just a cell suspension used to form biofilms. To prevent any confusion, the sentence is now modified (line 138-139 after revision in track change) and the preparation of cell suspension is now described under a separate section, ‘Preparation of cell suspension’ (line 134-139 after revision in track change) from the section of ‘Enzymes used in susceptibility test’.

5. More advanced technology should be applied to understand the structure of colony (biofilm?) before and after enzyme applicaitn. Not only the appearacne. For example, TEM for the inner structure; Staining for protein distribution.

-> Thank you for your comments. We used confocal laser scanning microscopy (CLSM) to image the biofilms exposed to enzymes and also to differentiate protein, carbohydrate, and cells in biofilms using fluorescent dyes. Those images are provided in Figure 3 and 8.

6. Why the extracellular DNA was secreted? Should explain this.

-> Thank you for the good comments. We tried to explain the role of extracellular DNA secreted in discussion section as in line 489-492 after revision in track change.

Reviewer #2: Line 38 "In this study, B. cereus strains from tatsoi and the ATCC 10987 reference strain" it's better to add codes of different strains used in this study like "B. cereus strains from tatsoi (BC4, BC10 and BC72)...........".

-> Thank you for your comments. We made a correction as the reviewer pointed out as in line 40 after revision in track change.

It is not clear how the isolated B. cereus strains were identified and confirmed as B. cereus. Chemical method or ribotyping? Please add results.

-> Thank you for your comments. As the reviewer pointed out, method for the identification and confirmation of B. cereus isolates was added in the materials and methods section of the revised manuscript. During the revision, we’ve found mistake in reference #18 (initial submission) and have changed as correct one (reference #18 after revision in track change). The added sentences on the identification and confirmation of B. cereus isolates were as follow (Line 99 - 110 in the revised manuscript after track change): To isolate B. cereus and B. thuringiensis, Polymerase chain reaction (PCR) amplification of the crystal (cry) gene was carried out for the biochemically identified B. cereus group isolates [Reference #19]. Biochemical identification for the B. cereus group was conducted by using the Vitek-II system with the BCL card (bioM´erieux, Inc., Marcy l’Etoile, France), according to the manufacturer’s directions. The specific primer pair K3 (5’-GCTGTGACACGAAGGATATAGCCAC-3’) and K5 (5’-AGGACCAGGATTTACAGGAGG-3’) was used for the identification of the cry gene (1,600 to 1,700 bp) for B. thuringiensis. Template DNA was preheated at 94℃ for 7 min. Then, the DNA was denatured at 94℃ for 60 s and annealed to primers at 58℃ for 90 s; and the PCR products were extended at 72℃ for 60 s for 30 cycles for the cry gene (ref #19, ref #20).

Line 123-124: Specify the staining solution?

-> Thank you for your comments. We removed CV which is an abbreviated form of crystal violet and mentioned the full name with more detailed description in the text to make it clear as in line 161-162 after revision in track change.

Which device was used for imaging Biofilm morphology?

-> Thank you for your comments. The images were captured by a commercial digital camera (EX2F, Samsung, Korea) and this information is added to the text as in line 217-218 after revision in track change.

Study Biofilm inhibition lacks reference/s. Authors are advised to mention reference clearly which was used for this study.

-> Thank you for your comments. We added the reference (Ref. 23) we followed to study biofilm inhibition and removal as in line 154-155 after revision in track change.

Line 197 "and 1 environmental isolate (strain BC72), strains BC4 and" replace "," with full stop and make two sentences.

-> Thank you for your comments. We made a correction according to the reviewer’s comment as in line 267 after revision in track change.

Line 377-389 "eDNA plays an important role in biofilm formation................ B. cereus strains in the environment". Figure 2 suggests that BC4 is susceptible to DNAse I, It is not clear why eDNA was not detected in biofilm of BC4? Suggest plausible reason.

-> Thank you for the good comments. Unfortunately, we do not have any clue why we are not able to detect eDNA of BC4 biofilm by running purified DNA on agarose gel. We just assume that eDNA of BC4 could be low molecular weight with a long range of size and may not be visible on the gel. However, it is still interesting that any DNA molecules not protected by impermeable membranes, which look red, seems to be degraded by DNase I as shown in Figure 3. It might suggest that eDNA could exist in BC4 biofilm.

It seems something was wrong with the data in figure 2C. The absorbance of crystal violet in the absence of degrading enzyme (-) (fig2C) is not comparable with the corresponding absorbance of crystal violet in fig 1A. Please comment.

-> Thank you for your point out. Currently, we do not understand why such a discrepancy occurred. However, it is true that experiment-to-experiment variation exists in biofilm experiments. 

Line 233-234: Why some enzymes promoted biofilm formation in a strain-dependent manner (Fig 2A and 2C).

-> Thank you for your comments. We hypothesize that in some cases the strains may not have the substances targeted by the exogenous enzymes or have the targeted substances protected. In addition, they may be able to degrade them by using the self-secreted protease enzymes. The degraded exogenous enzymes could serve as a nutrient and promote the biofilm formation. Therefore, the strain-dependency in promotion of biofilm formation could be due to strain-dependent variation in both EPS composition and self-secreted enzymes (ex. Staphylococcus aureus) (Kiedrowski et al., 2011). 

Careful English editing is advised, help from a native-language expert or paid services shall be taken.

-> Thank you for your advice. Our manuscript is edited by an English editing service. The editing certificate is uploaded. 

The word "in addition" in line 74 shall be replaced by an alternative word as it was previously mentioned in line 46 and then in line 82 again.

-> Thank you for your comments. The word “in addition” in line 74 has been changed to the word “Particularly”, considering the context, as in line 76-77 after revision in track change. 

Line 76: Word “although” is not required.

-> Thank you for your comments. We removed the word “although” from the sentence as the reviewer pointed out.

Line 97-99 “Among the B. cereus .......... seeds and environmental samples” can be written as “Among the B. cereus isolates, ones with highest biofilm formation abilities were selected from each category i.e. strain BC4 from microgreen samples, BC10 from seeds of microgreens and BC72 form water samples were selected for current study”.

-> Thank you for your comments. As the reviewer suggested, we rephrased the sentence as in lines 127-133 after revision with track changes.

---

## [Decision Letter · Decision Letter 1]

2 Jun 2021

Strain variation in Bacillus cereus biofilms and their susceptibility to extracellular matrix-degrading enzymes

PONE-D-21-00279R1

Dear Dr. Kim,

We’re pleased to inform you that your manuscript has been judged scientifically suitable for publication and will be formally accepted for publication once it meets all outstanding technical requirements.

Kind regards,

Muhammad Hussnain Siddique

Academic Editor

PLOS ONE

Additional Editor Comments (optional):

Reviewers' comments:

Reviewer's Responses to Questions

**Comments to the Author**

1. If the authors have adequately addressed your comments raised in a previous round of review and you feel that this manuscript is now acceptable for publication, you may indicate that here to bypass the “Comments to the Author” section, enter your conflict of interest statement in the “Confidential to Editor” section, and submit your "Accept" recommendation.

Reviewer #2: All comments have been addressed

2. Is the manuscript technically sound, and do the data support the conclusions?

Reviewer #2: (No Response)

3. Has the statistical analysis been performed appropriately and rigorously? 

Reviewer #2: (No Response)

4. Have the authors made all data underlying the findings in their manuscript fully available?

Reviewer #2: (No Response)

5. Is the manuscript presented in an intelligible fashion and written in standard English?

Reviewer #2: (No Response)

6. Review Comments to the Author

Reviewer #2: (No Response)

7. PLOS authors have the option to publish the peer review history of their article (what does this mean?). If published, this will include your full peer review and any attached files.

Reviewer #2: No

---

## [Editor Report · Acceptance letter]

7 Jun 2021

PONE-D-21-00279R1 

Strain variation in *Bacillus cereus* biofilms and their susceptibility to extracellular matrix-degrading enzymes 

Dear Dr. Kim:

I'm pleased to inform you that your manuscript has been deemed suitable for publication in PLOS ONE. Congratulations! Your manuscript is now with our production department. 

Kind regards, 

on behalf of

Dr. Muhammad Hussnain Siddique 

Academic Editor

PLOS ONE